# Standard and New Echocardio Techniques, Such as Global Longitudinal Strain, to Monitor the Impact of Diets on Cardiovascular Diseases and Heart Function

**DOI:** 10.3390/nu16101471

**Published:** 2024-05-13

**Authors:** Andrea Sartorio, Chiara Dal Pont, Simone Romano

**Affiliations:** Division of Internal Medicine C, Department of Internal Medicine, University of Verona, 37134 Verona, Italy; andrea.srt94@gmail.com (A.S.); dalpontc@gmail.com (C.D.P.)

**Keywords:** diet, echocardiography, cardiovascular disease, global longitudinal strain, hypertension, cardiovascular prevention, MASLD

## Abstract

“*The Seven Countries Study*”, published in 1984, was the first study to find a correlation between diet and mortality related to cardiovascular diseases (CVDs). Since then, many investigations have addressed the relationship between type of diet, or specific nutrients, and CVDs. Based on these findings, some traditional dietary models, such as the Mediterranean or Nordic diet, are recommended to prevent CVDs. Meanwhile, new diets have been proposed for optimal nutrition therapy, for example, the Dietary Approaches to Stop Hypertension (DASH) and the Mediterranean–DASH Intervention Diet for Neurodegenerative Delay (MIND). The main outcomes evaluated after implementing these dietary models are as follows: CVD-related death; the development of specific CVDs, such as myocardial infarction and hypertension; or biochemical parameters related to CVDs, i.e., non-HDL cholesterol, C-reactive protein (CPR) and homocysteine. However, the early impact of diet on heart functionality is less evaluated. Recently, the echographic measurement of left ventricle (LV) deformation by global longitudinal strain (GLS) has been introduced as a novel marker of clinical and subclinical cardiac dysfunction. This technology allows a subclinical evaluation of heart functionality since, differently from the traditional evaluation of left ventricle ejection fraction (LVEF), it is capable of detecting early myocardial dysfunction. In this review, we analyzed the available studies that correlate dietetic regimens to cardiovascular diseases, focusing on the relevance of LV strain to detect subclinical myocardial alteration related to diet. Evidence is presented that DASH and MIND can have a positive impact on heart functionality and that myocardial strain is useful for early detection of diet-related changes in cardiac function.

## 1. Introduction

Cardiovascular disease (CVD) is still the major cause of mortality worldwide, even with a declining incidence thanks to prevention (both primary and secondary) and treatments [1,2,3,4]. The main cause of CVD is atherosclerosis, a complex condition with a pathogenesis not yet fully understood that involves many factors [5]. Even with these limitations, many risk factors are known, both nonmodifiable (i.e., sex and genetics) and modifiable, such as smoking, high blood pressure (BP), apolipoprotein-B-containing lipoproteins, obesity, and diabetes mellitus (DM) [3,4]. Given these modifiable risk factors, the main intervention to reduce the incidence of CVD is to promote a healthy lifestyle, as proposed by the European Society of Cardiology (ESC) and American Heart Association (AHA) guidelines on CVD prevention, released in 2021 and 2019, respectively [3,4]. In these guidelines, diet is highlighted as a cornerstone of CVD prevention, as it can modify nearly all the modifiable risk factors (except smoking) [3,4,6,7,8,9]. Moreover, considering high BP as a pathology per sè, and not only a risk factor for CVD, lifestyle interventions can reduce BP by an average of 4–5 mmHg and increase the efficacy of pharmacologic therapy [10,11], which is of significant clinical relevance [12].

Many studies are focused on the selection of dietetic approaches to reduce CVD and hypertension and consequently mortality, starting from “*The Seven Countries Study*” to DASH (Dietary Approaches to Stop Hypertension) and MIND (Mediterranean–DASH Intervention for Neurodegenerative Delay) [13,14]. Moreover, diet impact is also crucial in metabolic dysfunction-associated steatotic liver disease (MASLD, previously known as nonalcoholic fatty liver disease—NAFLD) [15,16] and obesity [17]. 

Although mortality and morbidity are the classical endpoints of studies on diet, the availability of reliable biomarkers is helpful in monitoring the impact of diet on the pathogenetic mechanisms of CVD. In this respect, the application of new technologies during echocardiography can reveal subclinical myocardial alteration related to diet. Among these, left ventricle (LV) strain and global longitudinal strain (GLS) are currently at the forefront. In particular, GLS appears to be a more reproducible measurement, capable of giving a more comprehensive evaluation of myocardial function compared to the left ventricle ejection fraction (LVEF) [18,19,20]. In addition to GLS, which evaluates the longitudinal motion of the left ventricle in a two-dimensional (2D) manner, circumferential and radial strain can be obtained, both with a well-established clinical utility in various conditions [21,22,23]. In recent years, technologies that allow an evaluation of LV in three-dimensional (3D) or even three-dimensional (4D) have been released, with increasing evidence of their utility and feasibility [24,25,26]. The use of these technologies may be promising for future evaluation of diets’ impact on the heart. 

## 2. Discussion

### 2.1. The Atherosclerosis as Main Target of Dietetic Regimen

Atherosclerotic plaque formation is the cornerstone of CVD development. This is a complex phenomenon that involves low-density lipoprotein cholesterol (LDL-C), platelets, immune system cells, endothelium inflammation, arterial hypertension, and genetic factors [3,27]. Despite the complexity of this scenario and the wide number of components involved, risk factors related to diets were estimated to be responsible for 53% of CVD deaths [28]. The first factor identified as strongly related to the development of CVD and partially related to diet was cholesterol, and more precisely, LDL-C and other apo-B-containing lipoproteins [3,13]. Diet impact on LDL-C levels showed mixed results; in fact, some foods can lower LDL-C [29], and a meta-regression analysis showed a correlation between cholesterol in diet and total cholesterol in blood [30]. A similar correlation between cholesterol in diet and LDL-C was not found [30,31,32]. Thus, it is not surprising that in 2015, the Dietary Guidelines Advisory Committee did not recommend a stringent control of cholesterol in diet [33]. This is coherent with the notion that atherosclerosis is a multi-factor pathology and, therefore, to decrease CVD incidence, it is not sufficient to reduce the uptake of cholesterol (or even fat). In addition, many nutrients can exert a positive effect on the prevention of atherosclerosis, such as coenzyme Q10 (CoQ10), vitamin E, unsaturated fatty acids, and others [6,34]. Therefore, it is not surprising that all the major guidelines from cardiology society, both European and American, suggest a correct dietetic regimen instead of an elimination diet [3,4,30]. Therefore, we evaluated the cardiovascular effects of the four major dietetic regimes, which are summarized in Figure 1.

### 2.2. Dietetic Regimen

#### 2.2.1. Mediterranean Diet

The first and probably the most investigated diet is the Mediterranean diet, so termed because it recapitulates the eating habits of the societies that live in the Mediterranean basin. This diet is based mainly on vegetables, fruits, high-fiber products, and olive oil, with a limited intake of animal-derived products and saturated fatty acids [6,35]. A remarkable aspect of this diet is related to the potential anti-inflammatory and antioxidant actions exerted by several of its components, with a positive effect on atherosclerosis and endothelial function [36,37,38,39,40,41]. A comparison between the Mediterranean diet and a low-fat diet uncovered a lower incidence and mortality for CVD in people following the Mediterranean diet [42,43,44]. A meta-analysis of 49 studies conducted in 2019 found a modest benefit of the Mediterranean diet on CVD risk factors in primary prevention. Still, it was unable to define its impact on secondary prevention due to the limited number of studies [45]. Few studies have directly investigated the impact of the Mediterranean diet on echocardiography parameters. However, to the best of our knowledge, none of these studies used GLS. Considering traditional echocardiographic indices, a study conducted on 1937 adults showed that adherence to the Mediterranean diet was inversely associated with LV mass, even after adjusting this index with all possible confounders [46]. Similarly, a lower incidence of left ventricular hypertrophy (LVH) was observed in patients with higher adherence to the Mediterranean diet, although the male sex was associated with higher risk [47]. Only one study, conducted with magnetic resonance imaging (MRI) on 4497 adults, reported a positive association between the Mediterranean diet and LV mass. However, this was balanced by higher ejection fraction and stroke volume, so it was not indicative of adverse remodeling [48]. Other studies conducted on patients with chronic heart failure (CHF) revealed that patients more adherent to the Mediterranean diet had potential beneficial effects on biventricular systolic and diastolic function [49]. This last aspect is coherent with the observed protective effects of this diet on HF development [50].

We were unable to find reports about the use of GLS to evaluate the Mediterranean diet’s impact on the heart. We then searched whether a high-fat diet (HFD), in contrast to the Mediterranean diet, which is a low-fat diet, could impair ventricular function in preclinical settings. In a study conducted on 33 mice, an early LV dysfunction was detectable following feeding with HFD. This alteration was detected before the changes in the LVEF [51]. Another study showed similar findings, with a significant decrease of GLS significantly in mice on HFD, with a still normal LVEF [52].

#### 2.2.2. Dietary Approaches to Stop Hypertension (DASH)

The DASH trial, conducted in 1997, showed that a diet with a high intake of fruits, vegetables, low-fat dairy products, and a limited intake of salt and total fat was associated with lower BP [53]. The positive effect on BP was subsequently confirmed by various clinical trials and meta-analyses [14,54,55,56]. Similarly to the Mediterranean diet, DASH seems to protect against HF development [50]. Moreover, this diet was associated with lower LDL-C, HbA1c, and body weight. Another study confirmed a decreased incidence of CVD, coronary heart disease, stroke, and diabetes [57]. Effects of DASH on cardiac functionality were evaluated by a cross-sectional study on 4506 adults free of CVD conducted with an MRI. This study showed that a higher adherence to DASH (assessed with a validated food-frequency questionnaire) was related to a better end-diastolic volume and stroke volume [58]. The positive impact of DASH on traditional echocardiographic indices was proved in a small prospective study conducted on 13 patients with stable heart failure with preserved ejection fraction (HFpEF). In this cohort, after 25 days of DASH, a significant improvement in LVEF and stroke volume was denoted [59]. New echocardiographic indices were evaluated in a longitudinal study on 4651 adults from a community-based cohort that were re-evaluated 24 years after enrolment. At the final evaluation, patients with a higher adherence to the DASH diet showed a higher absolute value of GLS, while no changes were observed in the LVEF [60]. This finding is particularly interesting for a couple of reasons: first, a higher adherence to DASH in mid-life is associated with a higher absolute value of GLS in elderly life; second, GLS can detect a positive impact of DASH before the improvement of other widely used echocardiographic indices such as the LVEF. These aspects are summarized in Figure 2.

#### 2.2.3. Plant-Based Diets

Vegetarian (plant-based but with dairy products and eggs) and vegan diets (completely plant-based) have become popular in the last few years due to the proposed positive impact both on human health and the environment [61,62]. ESC and AHA guidelines currently support dietetic regimes that emphasize vegetables, as mentioned before. However, they do not recommend a completely plant-based diet [3,4]. The impact of a plant-based diet on CVD incidence and mortality was evaluated by recent meta-analyses that found a lower incidence of CVD, cancer, type 2 diabetes (T2D), and all-cause mortality in people who followed this diet, although with a high heterogeneity across studies [63]. However, an interesting aspect was outlined in a large prospective study (126,394 adults on a plant-based diet) included in this meta-analysis. In this cohort, only those who followed a healthful plant-based diet (e.g., with lower consumption of refined grains, fruit juice, potatoes, and animal-derived products) showed CVD and total mortality risk reduction; a unhealthful-based diet was instead associated with an increased risk of CVD, mortality, and cancer [62]. Fewer data are available regarding the vegan diet, with no clear evidence of CVD reduction, despite a positive impact on LDL-C and diabetes risk [64,65,66,67]. To the best of our knowledge, only two studies have evaluated the impact of plant-based diets on heart functionality. In a prospective study conducted on 456 individuals, with a median follow-up of 12.9 years, a lower risk of left ventricular diastolic dysfunction (LVDD), evaluated with echocardiography, was observed among people with a higher legume and vegetable intake [68]. Notably, in this study, even with low statistical significance, total and animal proteins were associated with higher risks of LVDD in women but not in men. In contrast, fresh vegetable consumption conferred higher protective effects in men than in women. No differences were observed between different ethnicities [68]. The other study compared vegan and omnivorous amateur athletes (22 vegan athletes and 30 omnivorous) and showed that vegan athletes had a lower left ventricular mass and relative wall thickness but also a higher GLS [69]. This study, despite its limited number of subjects, is the first that showed a positive impact of a plant-based diet on GLS.

#### 2.2.4. Mediterranean–DASH Intervention Diet for Neurodegenerative Delay (MIND)

Cognitive decline, and even dementia, share with CVD many risk factors, for instance, high BP, high cholesterol levels, and obesity [70,71,72,73]. Given this common background, various studies evaluated if the Mediterranean diet and DASH could also have a positive impact on cognitive decline. Meta-analyses found that the Mediterranean diet was associated with a reduced risk of cognitive impairment and neurodegenerative diseases [74,75,76]. Fewer data are available for DASH, although a positive impact on cognitive decline was observed [70,76]. Taking into account the positive effects of the Mediterranean diet and DASH, a new diet termed MIND (Mediterranean–DASH Diet Intervention for Neurodegenerative Delay), specifically formulated to prevent dementia and based on the previous, was proposed in 2015 [77]. This diet combines elements of the Mediterranean diet and DASH, with a major focus on green leafy vegetables and berries as preferred fruits [77]. The MIND diet, as expected, has shown to be effective in reducing the risk of cognitive decline, dementia, and even Alzheimer’s disease [35,76,77]. Considering that the MIND diet is based on the Mediterranean and DASH diets, its positive effect on CVD risk reduction is not surprising [78]. Only a single study directly evaluated the impact of the MIND diet impact on the heart, using echocardiography on 2512 adults from the Framingham Offspring Cohort [79]. Patients were evaluated both with traditional echocardiographic indices and with ventricular strain. The MIND diet was positively associated with LV mass and with LV diastolic functionality (represented by an E/e’ ratio), even if this last result was not confirmed after additional adjustment for cardiovascular risk factors. The association with LV mass does not appear to be related to adverse remodeling; in fact, no association was found with LV hypertrophy (LVH). Considering instead ventricular strain, MIND was inversely associated with GLS (not confirmed after adjustment) and global circumferential strain (GCS). Studies evaluating myocardial functionality in diets are reported in Table 1.

### 2.3. Cardiac Evaluation Regarding Diet in MASLD and Cancer

Particular attention must be placed on metabolic dysfunction-associated steatotic liver disease (MASLD, formerly known as nonalcoholic fatty liver disease—NAFLD). MASLD is the prevalent cause of liver-related morbidity in adults, with an increasing prevalence paralleled by the increase in obesity [15,16,80,81]. In MASLD, steatosis and subsequential hepatic inflammation can lead to fibrotic progression [80,82,83]. Mortality in patients with MASLD is mainly related to CVDs. However, patients with higher stages of fibrosis are at increased risk of liver-related mortality [80,84]. Notably, the CARDIA study demonstrated a positive correlation between NAFLD and impaired myocardial strain, independently of established heart failure risk factors [85]. Although guidelines from different scientific associations highlight lifestyle intervention as the first line of treatment for MASLD, underlining the importance of weight loss, specific diet recommendations are divergent [83,86,87,88]. The Mediterranean diet is the most recommended by scientific societies for MASLD treatment [83,86,87,88]. It is indeed associated with a low prevalence of cardiovascular disease, metabolic syndrome and type 2 diabetes, and liver damage [89,90,91]. Other dietary strategies, such as the low-fat diet, the vegetarian diet, or the intermittent diet, could be alternatives for treating NAFLD, as demonstrated by observational and prospective studies [92]. Although several studies demonstrate the positive impact of weight loss and the Mediterranean diet on cardiovascular disease and MASLD, there is no evidence of the correlation between lifestyle intervention and LV strain in NAFLD patients.

Another particularly relevant condition in which diet plays a role is cancer. Diet and food’s role in cancer development is a topic that has been studied for decades, and nowadays, diet and obesity are considered important risk factors for some types of cancers [93]. Various studies found a correlation between certain foods and an increase in some types of cancer (i.e., salted food and stomach cancer [94]). In contrast, other types of food have been associated with risk reduction (i.e., soy and prostate cancer [95]) [93]. The Western diet has been linked to an increased risk of obesity and cancer [96]. In contrast, other diets, like the Mediterranean, DASH, and plant-based, were instead associated with reduced risk [63,97,98,99]. Along these lines, intentional weight loss and a high intake of fruit and vegetables can reduce the risk of cancer recurrence and mortality [100]. On the other hand, diet can mitigate the overall impact of antineoplastic treatments. For instance, the prophylactic administration of flaxseed and its active components showed that it was partially protective against doxorubicin and trastuzumab-mediated cardiotoxicity, characterized by LVEF decrease, in mice [101]. On the other hand, resveratrol was capable of reducing the adverse effects of doxorubicin on LV contractility in a murine model [102]. Finally, in a small cohort of cancer survivors, subjects who received inorganic dietary nitrate supplementation (that can be found in green leafy vegetables [103]), despite a similar LVEF, showed a better left ventricular strain rate during early diastole [104].

## 3. Conclusions

Diet has an important role in prevention, both primary and secondary, and management of various relevant diseases. However, the direct impact of dietetic regimes on the heart is not yet fully understood, likely because the standard echo parameters, as EF, detect late alterations. GLS seems to be a promising technology that can fulfill this gap, allowing a more complete understanding of LV functionality and revealing its early alteration. Notably, in the reported studies, no significant differences between gender and ethnicity were found. This can be important to further optimize diet as therapy, but also to reveal if patients are achieving the expected benefits with that specific diet.

## Figures and Tables

**Figure 1 nutrients-16-01471-f001:**
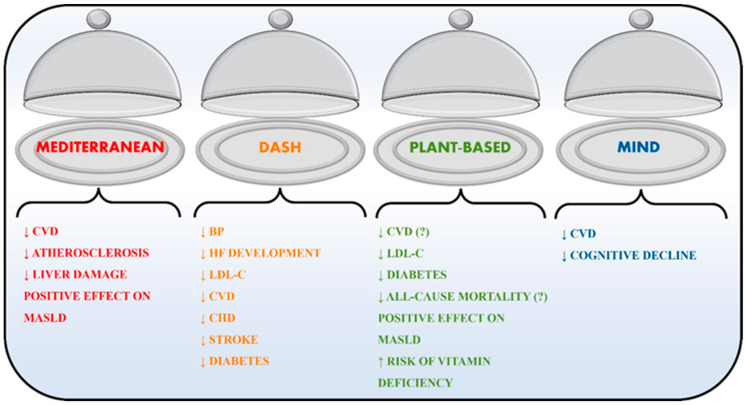
Cardiovascular effects of diets. All considered diets reduce the main outcomes, while this is uncertain for plant-based diets (all-cause mortality and cardiovascular disease). DASH: Dietary Approaches to Stop Hypertension; MIND: Mediterranean–DASH Intervention Diet for Neurodegenerative Delay; CVD: cardiovascular disease; MASLD: metabolic dysfunction-associated steatotic liver disease; BP: blood pressure; HF: heart failure; CHD: coronary heart disease, ?: uncertain Parts of the figure were drawn by using pictures from Servier Medical Art (smart.servier.com (accessed on 01/02/2024)). Servier Medical Art by Servier is licensed under a Creative Commons Attribution 3.0 Unported License (https://creativecommons.org/licenses/by/3.0/ (accessed on 01/02/2024)).

**Figure 2 nutrients-16-01471-f002:**
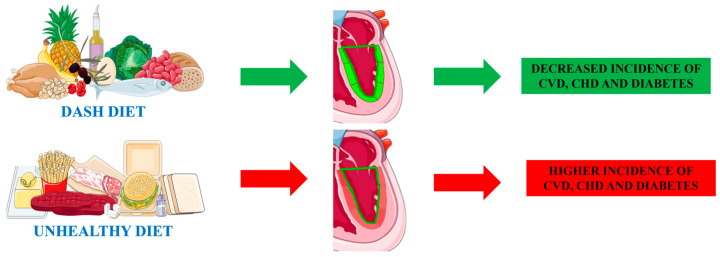
The cartoon explains the basic cardiovascular effects of a healthy diet (DASH) vs an unhealthy diet. The direct action of those diets on the heart can be assessed with global longitudinal strain (GLS). With this technology, early alteration of myocardial functionality can be assessed even if standard echocardiographic indices (such as ejection fraction, EF) are still normal. The green line delimiting the ventricle wall represents the EF that appears normal in both diets. The rectangles represent the GLS that is normal in the DASH diet (green), while it is altered in the unhealthy diet (red). DASH: Dietary Approaches to Stop Hypertension; CHD: coronary heart disease; CVD: cardiovascular disease. Parts of the figure were drawn by using images from Servier Medical Art (smart.servier.com (accessed on 01/02/2024)). Servier Medical Art by Servier is licensed under a Creative Commons Attribution 3.0 Unported License (https://creativecommons.org/licenses/by/3.0/ (accessed on 01/02/2024)).

**Table 1 nutrients-16-01471-t001:** Cardiac effects of dietetic regimes.

Author	Year	Participants	Study Design	Dietetic Regimen	Observed Effect on Heart of Considered Diet	Strain	LVEF
Gardener, H. et al. [46]	2015	1937 Adults >40 years old, without IS	cross-sectional	Mediterranean diet	lower LV mass	NE	NE
Bacharaki, D. et al. [47]	2022	127 Adults in dialysis	cross-sectional	Mediterranean diet	lower incidence of LVH	NE	NE
Levitan, E.B. et al. [48]	2016	4497 Adults 45–84 years old, without CVD	cross-sectional	Mediterranean diet	higher LV mass, LVEF and stroke volume	NE	Y
Chrysohoou, C. et al. [49]	2012	372 Adults with HFrEF	cross-sectional	Mediterranean diet	better left ventricular filling pattern	NE	N
Nguyen, H.T. et al. [58]	2012	4506 Adults 45–84 years old, without CVD	cross-sectional	DASH	better end-diastolic volume and stroke volume	NE	N
Hummel, S.L. et al. [59]	2013	13 Adults with hypertension and HFpEF	prospective cohort	DASH	better LVEF and stroke volume	NE	Y
Yi, S.Y. et al. [60]	2021	4651 Adults 45–64 years old	prospective cohort	DASH	higher longitudinal strain lower left ventricle mean wall thickness	Y	N
Razavi, A.C. et al. [68]	2020	456 Adults and children with preserved EF	prospective cohort	Plant-based diet	lower risk LVDD	NE	N
Król, W. et al. [69]	2020	22 Adult amateur runners	case-control	Vegan diet	lower LV mass, relative wall thickness and higher GLS	Y	NE
Walker, M.E. et al. [79]	2021	2512 Adults	prospective cohort	MIND	higher LV mass but not LVH. Higher LEVF and better GLS and GCS. After adjustment only better GCS was confirmed	Y *	N *

LVEF: left ventricle ejection fraction; GLS: global longitudinal strain; NE: not evaluated; Y: statistically significant improvement; N: no statistically significant improvement; LV: left ventricle; IS: ischemic stroke; CVD: cardiovascular disease; HFrEF: heart failure with reduced ejection fraction; DASH: Dietary Approaches to Stop Hypertension; HFpEF: heart failure with preserved ejection fraction; LVH: left ventricle hypertrophy; EF: ejection fraction; LVDD: left ventricular diastolic dysfunction; MIND: Mediterranean–DASH Intervention Diet for Neurodegenerative Delay; GCS: global circumferential strain. *: statistically significant alteration of LEVF, GLS, and GCS. However, after adjustment for covariates (systolic blood pressure, anti-hypertensive medication, smoking status, diabetes mellitus, total cholesterol to HDL-cholesterol ratio, BMI, ventricular rate, and physical activity), only a better GCS was confirmed.

## Data Availability

The original contributions presented in the study are included in the article, further inquiries can be directed to the corresponding author/s.

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
