# Peer review of "Standard and New Echocardio Techniques, Such as Global Longitudinal Strain, to Monitor the Impact of Diets on Cardiovascular Diseases and Heart Function"

_nutrients, 2024, doi:10.3390/nu16101471_

Round 1
Reviewer 1 Report
Comments and Suggestions for Authors
The review is dedicated to update the aspects of the link between the diets and the cardiovascular diseases investigated by advanced echocardiography techniques.The paper is innovative and the subject is original especially in consideration of the larger use of the GLS in many metabolic chronic diseases. The authors should report if other deformations parameters, like twist are normaly used in case to detect any myocardial damage suspected on the basis of uncorrect diet. Particularly for strain analysis could be of interest to add some information regarding the use of standard GLS or 4DStrain as a reconstruction of all the myocardial segments . The authors should add some other data if avaliable, regarding the eventual gender differences
Author Response
Reviewer 1
- The authors should report if other deformations parameters, like twist are normaly used in case to detect any myocardial damage suspected on the basis of uncorrect diet. Particularly for strain analysis could be of interest to add some information regarding the use of standard GLS or 4DStrain as a reconstruction of all the myocardial segments
We thank the reviewer for this suggestion. Unfortunately, we do not found studies in which these technologies were used to evaluate diets’ impact on heart. However, we added in the introduction section a brief description about other types of strain and 3D and 4D echocardiography.
- The authors should add some other data if available, regarding the eventual gender differences
We thank the reviewer for this suggestion. We included throughout the text this information in studies where differences were available (row 116, 121, 186-190). An additional sentence was also added in the conclusions.

Reviewer 2 Report
Comments and Suggestions for Authors
This is a brief review summarizing available evidence on the associations between various diet regimens and CVD risk with a focus on the relevance of echographic measurements of left ventricle deformation (global longitudinal strain) and cardiac functionality. Regardless of the limited number of studies available and reviewed here, overall the report is well written and the content appears to be useful and interesting to the readers. Below are some comments and suggestions.
Table 1: Consider participants vs patients: it could well be that not all of the individuals participated in these studies were patients (some are recruited from the general population, free of CVD). So it would make more sense if they were described as participants. In addition, it would help readers if a brief description of participant characteristics (i.e., who were these people?) was added to the Table (perhaps in a new column or added to the column with sample size). There is a type in the Table footnote: LEVF should be LVEF. In the Table footnote, the authors might consider listing covariates that were adjusted for (line 205).
Figure 2: The writing inside the heart image(s) is unreadable. I assumed there were some numbers/ percentages. The figure lacks a legend; a figure legend that clearly describes what is shown in the figure, including an explanation for the differently colored markings inside the heart images (green vs red), would be helpful.
Please explain what you mean by “a better…” when you describe GLS in relation to DASH diet (pages 141-145).
Please place ref #73 right after the study was initially mentioned, i.e., line 189-190 (after the name “Framingham Offspring Cohort”). Currently, it comes a few sentences later at the end of the paragraph, making readers wonder where the reference is.
I am not sure if the section on “MASLD and diet” is relevant to this particular topic of this review. As the authors themselves note there is no evidence/study on cardiac functionally (LV strain) and diet in MASLD patients. Instead, I wonder if it would be of interest to the readers if studies in the “diet and cancer” field evaluating cardiac functionality (LVEF, GLS, etc.,) were included.
I also wonder if the authors could comment on sex-specific findings, differences between men and women, and/or ethnicity/race (populations), if there are any. Perhaps also emphasize the importance of these aspects.
Comments on the Quality of English Language
It is well written manuscript; some minor checks are required.
Author Response
Reviewer 2
- Table 1: Consider participants vs patients: it could well be that not all of the individuals participated in these studies were patients (some are recruited from the general population, free of CVD). So it would make more sense if they were described as participants. In addition, it would help readers if a brief description of participant characteristics (i.e., who were these people?) was added to the Table (perhaps in a new column or added to the column with sample size). There is a type in the Table footnote: LEVF should be LVEF. In the Table footnote, the authors might consider listing covariates that were adjusted for (line 205).
Thanks very much for your comment. Table 1 has been edited according to your suggestions.
- Figure 2: The writing inside the heart image(s) is unreadable. I assumed there were some numbers/ percentages. The figure lacks a legend; a figure legend that clearly describes what is shown in the figure, including an explanation for the differently colored markings inside the heart images (green vs red), would be helpful.
Legend for Figure 2 was present in our draft but not in the final version, probably due to a technical error. We apologize for the inconvenience. We restored and edited the legend accordingly to your suggestions. We also removed the unreadable numbers from the image.
- Please explain what you mean by “a better…” when you describe GLS in relation to DASH diet (pages 141-145).
Thank you. We edited the sentences.
- Please place ref #73 right after the study was initially mentioned, i.e., line 189-190 (after the name “Framingham Offspring Cohort”). Currently, it comes a few sentences later at the end of the paragraph, making readers wonder where the reference is
Thank you very much for your suggestion. We moved the references.
- I am not sure if the section on “MASLD and diet” is relevant to this particular topic of this review. As the authors themselves note there is no evidence/study on cardiac functionally (LV strain) and diet in MASLD patients. Instead, I wonder if it would be of interest to the readers if studies in the “diet and cancer” field evaluating cardiac functionality (LVEF, GLS, etc.,) were included.
Thank you very much for your suggestion. As requested by the third reviewer, we have shortened the paragraph. We have also added some data regarding diet and cancer.
- I also wonder if the authors could comment on sex-specific findings, differences between men and women, and/or ethnicity/race (populations), if there are any. Perhaps also emphasize the importance of these aspects.
We thank the reviewer for this suggestion. We included throughout the text these informations in studies where differences were available (row 116, 185-189). An additional sentence was also added in the conclusions.

Reviewer 3 Report
Comments and Suggestions for Authors
In this paper , the authors reviewed the available data on new and standard echocardiographic techniques in order to monitor the effects of diet on cardiovascular function and heart disease.
ABSTRACT :
● The title is appropriate for the content of the manuscript.
The abstract is sufficiently informative and is consistent with the content of the paper. The authors describe the background in a brief but exhaustive way.
INTRODUCTION
● In the introduction the authors take adequate account of the existing knowledge.
DISCUSSION
● This section describes with sufficient detail the available data. The text and the references are consistent with the information presented in the paper. Tables and figures are appropriate .
Minor Comments
Paragraph 2.1 Row 71-72. The sentence “a meta-regression analysis showed…cholesterol in blood” should be referenced.
Paragraph 2.2.1 “Dietary..DASH”. Typo in paragraph numbering? It should be Paragraph 2.2.2
Paragraph 2.2.3 Row 169 Please note that the number of vegan athletes is 22, because 8 members of study group were excluded leaving n=22 participants.
Table 1 on page 6. Penultimate Row (Krol reference): see above comment
Last but not least paragraph "Diets in MASLD" should be somewhat shortened/changed/deepened to be more aligned to the manuscript’s title and other paragraphs contents.
Comments on the Quality of English Language
The english style is fine amd requires minor adjustment and typos correction
Author Response
Reviewer 3
- Paragraph 2.1 Row 71-72. The sentence “a meta-regression analysis showed…cholesterol in blood” should be referenced.
Thank you. We added the reference.
- Paragraph 2.2.1 “Dietary..DASH”. Typo in paragraph numbering? It should be Paragraph 2.2.2
We corrected the type; we apologize for that.
- Paragraph 2.2.3 Row 169 Please note that the number of vegan athletes is 22, because 8 members of study group were excluded leaving n=22 participants.
We corrected the error; we apologize for that.
- Table 1 on page 6. Penultimate Row (Krol reference): see above comment
We corrected the error; we apologize for that.
- Last but not least paragraph "Diets in MASLD" should be somewhat shortened/changed/deepened to be more aligned to the manuscript’s title and other paragraphs contents
We thank you very much for your suggestion. We shortened the part related to MASLD and, in agreement with the second reviewer, we added a part related to cancer and diet.
